# UAV-Based Hyperspectral Monitoring Using Push-Broom and Snapshot Sensors: A Multisite Assessment for Precision Viticulture Applications

**DOI:** 10.3390/s22176574

**Published:** 2022-08-31

**Authors:** Joaquim J. Sousa, Piero Toscano, Alessandro Matese, Salvatore Filippo Di Gennaro, Andrea Berton, Matteo Gatti, Stefano Poni, Luís Pádua, Jonáš Hruška, Raul Morais, Emanuel Peres

**Affiliations:** 1Engineering Department, School of Science and Technology, University of Trás-os-Montes e Alto Douro, 5000-801 Vila Real, Portugal; 2Centre for Robotics in Industry and Intelligent Systems (CRIIS), INESC Technology and Science (INESCTEC), 4200-465 Porto, Portugal; 3Institute of BioEconomy, National Research Council (CNR-IBE), Via G. Caproni, 8, 50145 Florence, Italy; 4Institute of Geosciences and Earth Resources, National Research Council (CNR-IGG), Via Moruzzi 1, 56124 Pisa, Italy; 5Department of Sustainable Crop Production (DI.PRO.VE.S.), Università Cattolica del Sacro Cuore, Via E. Parmense 84, 29122 Piacenza, Italy; 6Centre for the Research and Technology of Agro-Environmental and Biological Sciences, University of Trás-os-Montes e Alto Douro, 5000-801 Vila Real, Portugal

**Keywords:** remote sensing, imaging sensor, radiometric calibration, unmanned aerial vehicles, bands co-registration, hyperspectral data cube

## Abstract

Hyperspectral aerial imagery is becoming increasingly available due to both technology evolution and a somewhat affordable price tag. However, selecting a proper UAV + hyperspectral sensor combo to use in specific contexts is still challenging and lacks proper documental support. While selecting an UAV is more straightforward as it mostly relates with sensor compatibility, autonomy, reliability and cost, a hyperspectral sensor has much more to be considered. This note provides an assessment of two hyperspectral sensors (push-broom and snapshot) regarding practicality and suitability, within a precision viticulture context. The aim is to provide researchers, agronomists, winegrowers and UAV pilots with dependable data collection protocols and methods, enabling them to achieve faster processing techniques and helping to integrate multiple data sources. Furthermore, both the benefits and drawbacks of using each technology within a precision viticulture context are also highlighted. Hyperspectral sensors, UAVs, flight operations, and the processing methodology for each imaging type’ datasets are presented through a qualitative and quantitative analysis. For this purpose, four vineyards in two countries were selected as case studies. This supports the extrapolation of both advantages and issues related with the two types of hyperspectral sensors used, in different contexts. Sensors’ performance was compared through the evaluation of field operations complexity, processing time and qualitative accuracy of the results, namely the quality of the generated hyperspectral mosaics. The results shown an overall excellent geometrical quality, with no distortions or overlapping faults for both technologies, using the proposed mosaicking process and reconstruction. By resorting to the multi-site assessment, the qualitative and quantitative exchange of information throughout the UAV hyperspectral community is facilitated. In addition, all the major benefits and drawbacks of each hyperspectral sensor regarding its operation and data features are identified. Lastly, the operational complexity in the context of precision agriculture is also presented.

## 1. Introduction

Satellite-based remote sensing sustained a major shift while embracing spectral imaging technology [1]. Indeed, this enabled the acquisition of multispectral images, promising to areas such as mining and geology [2], agriculture [3,4,5,6], environment [7,8], research on temporal evolution/changes [9,10], security and military agencies [11]. The Earth Resource Technology Satellite (ERTS) program—later called Landsat—has been using space sensors of medium resolution (tens of meters) to acquire multispectral data [12] since the 1960s. More recently, SPOT satellites (Satelitte Pour l’Observation de la Terre) have been another important source of high-quality, medium-resolution multispectral data [12]. Since then, technological developments have allowed space sensors’ performance to be compared to that from digital aerial cameras used in manned aircrafts data [12,13]. However, important differences remain, strictly related to each platform characteristics. While global access is an obvious advantage of satellites over manned aircraft, space sensors began to be developed many years before launch, which makes it harder to have reflected upon them technological advances that could meanwhile have emerged. Manned aircrafts are more flexible and can be used if weather conditions are favorable. Up until recently, only these remote sensing platforms allowed for the acquisition of spectral images, mainly due to the equipment’s size and weight [14,15]. However, the high-cost of commercial satellite images, which only provide spatial resolution of up to a few meters [14], considered coarse for some applications, despite being possible to conduct classification applications [16]. Furthermore, costs associated with manned flights’ logistics make these platforms unattractive for many applications, particularly those who require high- spatial and/or temporal resolutions.

The last decade has seen significant developments both in unmanned aerial vehicles (UAVs) and in associated technologies, as depicted in Figure 1. Improved sensors’ quality, increased payload capacity, expanded flight capabilities, reduced sensors’ dimensions, better data processing capabilities, are some of the most noteworthy advances achieved. UAV-based remote sensing is since becoming more accessible, cost-effective, and adaptable to different scenarios (crops, terrains’ orography and size of the area to be analyzed).

Among other applications, UAVs are the more usual platform to acquire data on both agriculture and forestry applications. Data’ quality and detail largely rely on the coupled sensor spectral range. Traditional and low-cost optical sensors, visible (RGB) or near infrared (NIR), have proven their usefulness in many applications [18], specially where only low detail data is required [19]. Obviously, the size of the covered area is considerably smaller than that made possible by manned aircrafts or by satellites, however, the spatial and temporal resolutions are considerably higher [10,20]. Fixed-wing UAVs can provide coverage of several square kilometers per working day [15].

However, when the objective is to profile materials and organisms, these sensors’ accuracy and spectral range may not suffice. In fact, and depending on the application, it may be necessary to measure several regions of the electromagnetic spectrum with detail. Here enters spectral imaging.

Spectral imaging is a technique presented by Goetz et al. [21] that enables gathering an object’s spatial and spectral information. Originally proposed to be used in earth remote sensing applications, it is based on the principles of spectroscopy [22]. Indeed, this technique can use the visible spectrum (red, green and blue—RGB—as more common cameras), infrared, ultraviolet, x-rays or combinations of these bands [23]. Therefore, each spectral image pixel potentially has data from hundreds of bands across different regions of the electromagnetic spectrum. The success of spectral imaging can be measured by the many different methods and associated technologies proposed over the past few decades. From these, whiskbroom, push-broom, staring and snapshot are the most commonly used approaches in remote sensing [24].

Regardless the approach used to acquire spectral imaging, data can be represented through a three-dimensional hypercube: spatial information is represented in two dimensions (x, y) and the third dimension represents spectral information (λ). Typically, hyperspectral sensors define narrow bandwidths and obtain a few hundred spectral bands per image’ pixel. Thus, each pixel is formed by a specific spectral profile containing location data, followed by hundreds of digital numbers aligned with the corresponding spectral bands.

From the four approaches typically considered when acquiring hyperspectral imaging in remote sensing two—push-broom and snapshot—account for most readily available solutions and were therefore chosen to this study. Figure 2 illustrates the main principles behind both approaches.

The push-broom approach—also known as line scanning—has been used for a long time by some earth observation satellite systems. This method can simultaneously acquire a range of spatial information, as well as spectral information corresponding to each spatial point in the scanned range. Images with a spatial dimension (y-line) and a spectral dimension (λ) can be obtained at the same time with a set of inline-aligned charge coupled device (CCDs). In the last two decades, better optical designs, improvements in electronics, and advanced manufacturing have all contributed to improve CCDs’ performance. However, during that time, the underlying optical technology has not really changed. In fact, modified forms of the classic spectrometer layouts remain standard. This still is the basis of push-broom concept.

As shown in Figure 2a, the push-broom approach acquires images line by line, in which the spatial information is displayed along one axis and the wavelength information on the other. Then, the spectral image data cube can be obtained by scanning the strip in the direction of another spatial axis. Obviously, this approach implies that either the sensor or the object must be in motion. This movement should be synchronized with the acquisition rate to guarantee full coverage of the object (or area) of interest [12]. In this study, each entire line (or row) of the hyperspectral imagery is acquired at once, with the UAV’s motion providing the down-track scanning.

On the other hand, the snapshot (or single-shot) approach allows the recording of spatial and spectral information using a matrix detector. Snapshot spectral sensors use optical designs that differ greatly from standard forms, and in contrast with push-broom—which needs to scan in the spatial or spectral dimensions, limiting their temporal resolution—snapshot is an imaging technique without the need to scan. As shown in Figure 2b, snapshot enables the acquisition of a complete spectral data cube in a single integration, generating images directly from the areas of interest. This approach allows to directly acquire data, which reduces the post-processing complexity to obtain a 3D data cube. However, its spatial and spectral resolutions are limited, since the total number of voxels (regular grid in a three-dimensional space) cannot exceed the total number of pixels on the CCD. Therefore, for a given CCD, it is always possible to increase spatial sampling at the expense of spectral sampling and vice versa [26]. Moreover, this approach acquires hypercubes from different spectral ranges and number of bands and produces non-registered bands, especially when working coupled to an UAV. So, co-registration is needed to avoid band misalignment. This type of hyperspectral imaging approach is still evolving since average size areas can be measured at a relative few wavelengths, mostly in the wavelength range of 400–910 nm [21].

The advent of large-format (4 megapixel) detector arrays brought the capability to measure millions of voxels simultaneously. It is this large-scale measurement capacity that makes snapshot spectral imaging practical and useful. Indeed, it is only by making use of large arrays of detector elements that these advantages can be achieved. Furthermore, it is only in the past 10 years or so that spatial and spectral resolutions achieved by snapshot imaging systems have become sufficient so both devices are now commercially viable. Hyperspectral sensors, especially those covering wavelengths ranging from 400 nm to 1000 nm (visible + NIR), have evolved regarding both weight and dimension, enabling them to be coupled to UAVs. Table 1 summarizes the main hyperspectral sensors currently available on the market and capable of being transported by UAVs. A few works have focused on UAV-based hyperspectral imagery for agricultural and environmental monitoring [27,28]. Tommaselli et al. [29] evaluated the capability of different 2D changes in band registration of the time-consecutive camera images. The results by Honkavaara et al. [30] indicated that the band registration of such images feature-based matching and 2D image transformation provided good registration in flat agricultural scenarios. Jakob et al. [31] presented MEPHySTo as a toolbox for pre-processing UAV hyperspectral data consisting of a pre-processing chain fitted for challenging geometric and radiometric correction. Booysen et al. [32] employed UAVs with hyperspectral sensors to detect rare earth elements and this solution had the advantage of quick turn-around times (<1 d), low detection limits (<200 ppm for Nd) and is ideally suited to support exploration campaigns. Finally, Geipel et al. [33] investigated the potential of in-season airborne hyperspectral imaging for the calibration of robust forage yield and quality estimation models and the prediction performance of PPLSR models was the highest compared to simple linear regression (SLR) models, which were based on selected vegetation indices and plant height.

Headwall’s Nano Hyperspec (push-broom) and SENOP’s VIS-VNIR Snapshot hyperspectral sensors will be presented and evaluated in this study. Recent studies using hyperspectral push-broom sensors were able to contribute to the state-of-the-art in grapevine nutrient status assessment [34], determination of chlorophyll content in grass communities [35], estimation of soil moisture content in an arid region [36], improvement of grapevines pest surveillance [37], estimation of maize yield and flowering using multi-temporal data [38], combining LiDAR data for classification of mangrove species [39], and mapping of intertidal macroalgae [40]. With regard to snapshot hyperspectral sensors, they were used in precision viticulture [41] to assess grapevine biophysical parameters [42], for the detection and mapping citrus trees affected by phytosanitary issues [43], to estimate potassium in peach leaves [44], to classify plant communities using data from multiple sensors [45], for the estimation of barley and grass nitrogen and biomass [46], and to estimate different winter wheat crop parameters and yield [47,48]. The theoretical advantages and disadvantages of each of these hyperspectral imaging acquisition approaches are presented and discussed in the next section and will be verified through the processing and analysis of data acquired by the two sensors over similar study areas. This work will assist researchers, agronomists, winegrowers and UAV pilots not only to select the most appropriated hyperspectral sensor for their intended purposes, but also to have data collection protocols and methods to achieve faster processing techniques and integrate multiple data sources. As such, only facts based on obtained results will be conveyed to the readers, avoiding potentially biased opinions.

## 2. Materials and Methods

As previously established, this work main objective is to compare two hyperspectral sensors whose characteristics allow them to be coupled with a UAV. This will be carried out by considering the different stages that lead up to an orthorectified hyperspectral mosaic: flight planning and execution, data pre-processing—mainly the radiometric calibration operations [49,50,51]—and final processing. Each of these stages and the selected study areas are described in this section. Hereinafter all subsections will be organized by hyperspectral sensor and respective study areas to improve both the readability and the overall readers’ experience.

### 2.1. Sensors and Platforms

#### 2.1.1. Nano-Hyperspec and DJI M600

The DJI Matrice 600 (M600) Pro hexacopter was the flight platform coupled to Headwall’s Nano-Hyperspec sensor. To assure sensor’s stability, a Ronin-MX gimbal was mounted in the M600. A 12 mm lens with a horizontal field of view (FOV) of 21.1° was used. Each line of pixels comprises 640 spatial pixels and 272 spectral bands with a total of 172,800 pixels-waveband combinations. The sensor acquires 12-bit data, i.e., 4096 brightness levels across the visible and near-infrared spectrum, ranging from 400 to 1000 nm, with a sampling interval of 2.2 nm and a full width at half maximum (FWHM) of approximately 6 nm [52]. Five Global Positioning System (GPS) antennas are mounted on the M600 upper plate and opposite arms: three for the UAV’s navigation and altitude determination and two, mounted in the arms, used in the positioning of the hyperspectral sensor. An inertial measurement unit (IMU) is also used to account for the effects of roll, pitch, and yaw. With this equipment profile, the M600 has a flight autonomy of approximately 20 min. More sensor specifications are presented in Table 2 and the aforementioned components are depicted in Figure 3.

Nano-Hyperspec is a push-broom sensor, which means that it acquires radiometric information corresponding to each spatial line in the scanned range. When the light enters through the lens, the optics inside the sensor disperse the light in the spectral axis, acting similar to 640 spectrometers—size of CCD line. After entering through the lens, the light is blocked at the optical system’s entrance port. The part corresponding to a small slit is let in, ensuring that only the rays corresponding to the strip of the observed terrain will enter. This spectrograph is based on offner spectrometers that have been most widely developed for hyperspectral imaging because they have better image quality and spectral resolution than other types of spectrometers [53]. The optical system included in the Nano-Hyperspectral sensor is shown in Figure 4.

The slit size defines the spatial dimension of each scanned line, corresponding to the CCD array placed in the focal plane (Figure 4). The light that reaches each CCD is dispersed through the grating, originating the spectral dimension. Therefore, each pixel can be seen as a 3D data cube, as it has the spatial dimension (field of view, flight direction) and a spectral dimension (spectral axis, in this case 272 values). Figure 5 illustrates this process.

#### 2.1.2. Senop Hyperspectral HSC-2 and DJI M600

A DJI M600 was also used as the flight platform (Figure 6). Moreover, the requirements for the single shot sensor are similar to those from Nano-Hyperspec. It is also necessary for the sensor to be mounted on a stabilized gimbal to minimize the motion caused by the UAV movements. A self-developed payload system was used to acquire hyperspectral data with a SENOP HSC-2 hyperspectral camera, consisting of a camera support for both static at ground and UAV dynamic acquisition, a stabilized gimbal mounted with an adaptable support for different kind of UAV models.

Senop Hyperspectral sensor HSC-2 can be used as a standalone device. If this is the case, Senop HIS software or other compatible software is needed to initialize and control the sensor. Before the acquisition process, both a sequence and a script need to be created. While the sequence defines the optical parameters, i.e., the wavelengths to be measured, bandwidth selection and integration time (exposure), the script—composed of the measure interval, triggering mode, exposure, number and width of spectral bands—defines the measurement parameters, such as how many times the sequence is to be measured, how the recording is started, among others. The script needs then to be uploaded to the HSC-2 sensor.

This sensor was used to test the single shot hyperspectral data acquisition mode. It is characterized by a true global shutter snapshot sensor based on Fabry-Pérot Interferometer (FPI) technology and composed of two CMOS sensors that redirect the light rays by a beam splitting device and a beam splitting component (Figure 7).

Each sensor is responsible for a particular spectrum range and has only one channel to receive energy: one of the sensors has been optimized to record the visible bands (500 to 636 nm), while the other acquires longer visible and NIR (VNIR) bands (650 to 900 nm). The point where the sensors switch—636 nm wavelength—has a so called “gap-area” where data is invalid. The set of different spectral bands is formed through successive frame grabbing. Furthermore, the user can select the desired spectral bands, range limits and spectral resolution of the hyperspectral cubes to be acquired by the sensor.

The process by which spectral images are built when using FPI technology can result in an imperfect overlap between the cube’s bands, along the spatial axis. Aasen et al. [54] called this set of bands an unregistered band package. Sensors’ misalignment and possible variations in altitude during exposure may also affect bands’ displacement. Thus, an appropriated calibration design and a proper analysis of the calibration data are needed. Moreover, all pixels are true pixel: no interpolation is used. Table 3 shows Senop Hyperspectral HSC-2 specifications.

### 2.2. Radiometric Correction

Despite being carefully calibrated by manufacturers, hyperspectral sensors of any type may need recalibration due to variations between laboratory and flight conditions [55] or due to small shifts of the spectral parameters over time [56]. It is therefore considered a good practice to assess the performance and reliability of new products, as well as conducting periodical assessments of the sensors in use [57].

#### 2.2.1. Nano-Hyperspec

Radiometric correction of the Nano-Hyperspec sensor is carried out in two main steps: the dark current assessment for the area detector, responsible for the conversion of incident photons into electrons to quantify light intensity [58]; and a white reference assessment for the optical system, responsible to form the actual image and the spectrometer [59]. The former is carried out prior to each flight, by collecting a hypercube with the lens’ cap on, and the latter in two steps: (1) by pointing a white target to adjust the sensor exposure and frame period; and (2) deploying a tarp with three different gray reflection levels, provided by the manufacturer (Figure 8), and placed within the interest area, enabling a simple gain calibration to take place. Subsequently, DN counts are converted into radiance values using Headwall’s SpectralView software.

As previously stated, obtaining imagery in reflectance units is required for the comparison of disparate data sets, collected from different or the same areas, over time. For that purpose, at-sensor radiance needs to be converted to at-surface reflectance. Scene reflectance generation is achieved using the empirical line method (ELM) [27], which is a calibration procedure to derive surface reflectance from at-sensor radiance [58], that is assumed to be linear [59,60]. The conversion from radiance to reflectance through ELM is then assured using the different grey-scale calibration panels.

#### 2.2.2. Senop Hyperspectral HSC-2

A dark calibration is needed to determine the dark current of Senop Hyperspectral sensor. If it is triggered under completely light-free conditions, this noise becomes visible within the resulting image and can be subtracted from the acquired raw image DN. As the camera’s shutter cannot be manually closed, a plastic cap is used to cover the optics and then acquire the dark current signal. In this work, the image dark current subtraction was carried out using Matlab Hyperspectral Processing Workflow (MathWorks, Natick, MA, USA).

The first operation in the radiometric correction process consists in the conversion from DN to radiance, using the gains derived from factory calibration. The following step converts from radiance to reflectance using reference reflectance panels and the ELM [27]. In this work, the ELM was applied using three reference 100 × 50 cm OptoPolymer (OptoPolymer—Werner Sanftenberg, Munich, Germany) homogeneous and Lambertian surface panels, with 97%, 56% and 10% reflectance. These three panels were previously characterized using a Spectralon (Spectralon, Labsphere, North Sutton, NH, USA) reference 25 × 25 cm panel with 99% reflectance, and a GER spectroradiometer (Spectra Vista Corporation, Poughkeepsie, NY, USA) (Figure 9). Within the empirical line correction, reference spectra are compared to extracted image’ spectra of ground targets to estimate the correction factors for each band [61]. Finally, the hyperspectral mosaic is converted to surface reflectance by applying those correction factors.

### 2.3. Processing Workflow

#### 2.3.1. Nano-Hyperspec

Nano-Hyperspec processing workflow is shown in Figure 10 as implemented in Headwall’s Spectral View software (Headwall Photonics, Inc., Bolton, MA, USA). As it can be seen, raw cubes’ calibration is the first required step (section 1 of Figure 10). Sensor’s measurements are converted into radiance using measurements at the ground (dark current). Next, the reference target measurements are used to convert radiance into reflectance. This normalization allows easier data manipulation because values are within the 0 to 1 range (0% to 100%) and are consistent, regardless of the lighting conditions when data was acquired. The next step consists in the orthorectification process (section 2 of Figure 10), where individual cubes taken over an area during a single scan (flight) are combined. Usually, several scanlines are associated into frames (images or individual hyperspectral cubes) that will be stitched together, creating a hyperspectral swath. As mentioned in Section 2.1.1, scanline sensors are highly sensitive to motion and therefore a stabilized gimbal and an IMU are used. Every pitch, roll and yaw change automatically carried out to follow the pre-programmed flight plan is transcribed into the acquired data. For a given cube, several parameters are required to be set. Therefore, this is an iterative process, where a fit of orthorectified cubes is analyzed over Google Earth maps. Parameters need to be adjusted until a satisfactory result is achieved, namely pitch, yaw, roll and altitude offset. The latter is of crucial importance since it is related to the surveyed terrain altitude and may influence the orthorectification process. Thus, the suggestion is to start by optimizing this parameter over a cube where visible reference points—acting as ground control points (GCPs)—can be determined. The remainder parameters are then adjusted according to the acquired fit. A good strategy is to consider the elevation profile of the scanned area. Indeed, an area with no significant changes in altitude enables the setting of parameters for multiple cubes simultaneously (batch process), which can save a significant amount of processing time. However, the direction from where the cubes are captured needs to be considered, because for each direction the parameters act differently. For this reason, only interleaved strips can be considered to apply the same parameters. Furthermore, the Digital Elevation Model (DEM) of the scanned area plays an essential role in the orthorectification process. Thus, low-resolution DEM (e.g., Shuttle Radar Topography Mission—SRTM) that cannot capture the terrain diversity will acquire insufficient results for those areas. To prove DEM’s influence in the orthorectification process, three flight campaigns were conducted: a campaign over two steep slope areas with different flight orientations and a flat coastal area, which are detailed in Section 2.4.

#### 2.3.2. Senop HSC-2

The processing workflow method can be summarized as presented in Figure 11. Matlab was used to process the hypercube acquired by the HSC-2 sensor and for dark current subtraction, radiometric calibration, mosaic filtering, index calculation and to produce the classified map as the final product. Agisoft Metashape (AgiSoft LLC, St. Petersburg, Russia) was used for lens correction and single bands mosaicking. Finally, QGIS was used for multi-bands matching.

In detail, a first procedure—called Multi-Bands Matching—for bands co-registration was implemented in Matlab. Single spectral bands of one image are acquired with a small temporal difference. Depending on the speed, movement and vibrations of the aerial platform, this results in a spatial shift between the single bands of the data cube (Figure 12). The correction of this mismatch could be carried out using an image-matching algorithm. Feature-based registration techniques used by Matlab Registration Estimator App automatically detect distinct image features such as sharp corners, blobs, or regions of uniform intensity. The moving image undergoes a single global transformation to provide the best alignment of corresponding features with the fixed image. Speeded-Up Robust Features (SURF) algorithm detects blobs in images and supports changes in scale and rotation. These blob detection methods are aimed at finding regions in a digital image that differ in properties—such as brightness or color—when compared to their surrounding regions. Within this context, a blob can be defined as a region within an image in which some properties are constant or approximately constant, which means that all of its pixels can be deemed similar.

However, results from the first procedure were not sufficiently accurate for all bands in this work. To overcome this issue, a second procedure—called Single Band Mosaicking—aimed at processing the co-registration for each single frame using unsupervised feature—based registration techniques was implemented. This action needs a preliminary pre-processing step using Matlab, where each band was extracted from each frame, radiometrically corrected and stored for the consequent single band mosaicking using Agisoft Metashape. After that, a supervised procedure of georeferencing using GCPs was carried out in QGIS software.

A specific focus regarding the sensor’s calibration needs to be detailed. Senop HSC-2 is a hyperspectral snapshot sensor. Distortions caused by internal features, related mainly to both radial and tangential distortions, occur. Radial distortions are related to the shape of the lens and mostly become visible as a “barrel” or “fish-eye” effect. Tangential distortions can be caused by a non-parallel assembly of the lens with regard to the image’s plane. HSC-2 hyperspectral sensor lens distortion parameters were determined using Agisoft Lens tool, which uses a checkerboard pattern projected on a flat screen or printed out. At least five images need to be acquired from different angles and orientations. Using these images, the internal parameters and therefore the distortion coefficients can be calculated. The internal parameters can be expressed by the characteristic matrix, which includes focal length (fx and fy), skew and center coordinates (cx and cy). The distortion coefficient matrix comprises the radial distortion coefficients k1, k2, k3 and the tangential distortion coefficients p1 and p2. All those parameters were estimated for each single band and applied in the alignment process on Agisoft Metashape.

### 2.4. Study Areas

This study intends to review in detail the various stages underlying the use of each type of hyperspectral sensor, and to evaluate their overall performance regarding both operation and quality—geometric and radiometric—of the generated orthorectified data. This reason alone supports the selection of different areas in two countries, without risking any influence on both the conclusions and pertinence of this study. Tests were conducted in four vineyards—two Portuguese and two Italian—that are characterized for having different grapevine varieties, density and landscape. Notwithstanding, study areas of similar dimensions and topographies were selected. The Nano-Hyperspec sensor was used in the Portuguese study areas and the Senop Hyperspectral sensor HSC-2 was used in the Italian study areas.

A flight plan was designed prior to each field campaign, considering flight’s height, spatial resolution requirements, area to cover, and lighting conditions. Morning hours close to solar noon and under a clear sky are the best flight conditions to both avoid shadows and to ensure optimal atmospheric conditions. The Universal Ground Control Station [62] desktop application was used for the UAV flight planning and to control both sensors used in this study.

#### 2.4.1. Nano-Hyperspec

The first test area is a 1.5 ha vineyard parcel—190 m, in average, above sea level (ASL)—belonging to Quinta do Vallado, which is located in the city of Peso da Régua, well within the heart of the Douro Demarcated Region, in Portugal (Figure 13A). The terrain has an approximately constant slope of 20% and grapevines are spaced 2.2 m × 1.3 m (respectively, inter-row and intra-row), and trained to a vertical-shoot-positioned trellising. The second test area is a 2 ha vineyard parcel, 250 m ASL (in average), located in Lousada, within the “Vinhos Verdes” region, in Portugal (Figure 13B). The terrain is mostly flat, and the grapevines are spaced 2.8 m × 1.75 m (respectively, inter-row and intra-row).

To verify the relief/altitude effect in the orthorectified mosaics generated from the acquired hyperspectral swaths, a sea level test area was added to the study. For this purpose, a 5 ha coastal area (Figure 13C), located in Viana do Castelo, Portugal, was considered.

#### 2.4.2. Senop Hyperspectral Sensor HSC-2 Study Areas

The Italian study areas represent two vineyards characterized by different varieties, density, and landscape. The first test area is in Borgonovo Val Tidone, Piacenza, Italy (Figure 14D). The flight was conducted in 2020 in a 1 ha Barbera vineyard—215 m ASL—that was established in 2003 (hereinafter referred as “Piacenza”). It is owned by Tenuta La Pernice. Grapevines are spaced 2.4 m × 1 m and trained to a vertical-shoot-positioned trellising. As for the second test area, the flight was carried out in 2019 in a 1.2 ha vineyard—355 m ASL, in average—planted in 2008 (hereinafter referred as “Chianti”), owned by Castello di Fonterutoli farm and located near Castellina in Chianti, Siena, Italy (Figure 14E). Sangiovese cv. (*Vitis vinifera*) grapevines were trained with a vertical shoot-positioned trellis system and spur-pruned single cordon. The vine spacing was 2.2 m × 0.75 m (respectively, inter-row and intra-row) and the rows were NW-SE oriented, on a slight southern slope. A detailed description of the area can be found in Matese and Di Gennaro [19]. Table 4 presents the main characteristics of the five study areas used in this study.

### 2.5. Data Collection

A flight plan was designed prior to each field campaign, considering flight’s height, spatial resolution requirements, area to cover, and lighting conditions. Morning hours close to solar noon and under a clear sky are the best flight conditions to both avoid shadows and to ensure optimal atmospheric conditions. The Universal Ground Control Station [24] desktop application was used for the UAV flight planning and to control both sensors used in this study.

#### 2.5.1. Nano-Hyperspec

In the first study area—Vallado, in Figure 13A—the hyperspectral swaths were collected using the M600 with a 40% sidelap, at a speed of 2 m/s and a height of 32 m, scanning at a frame rate of 100 fps, to acquire an approximately square pixel. A total of 11 swaths were required to cover the area, with a 2 cm GSD. The total flight time was of 16 min and 30 s. For the Lousada study area—in Figure 13B—a total of 13 hyperspectral swaths were scanned, reaching a GSD of 3 cm with a 40% sidelap, flying at a speed of 3 m/s, at a height of 48.6 m above the ground, and scanning at a frame rate of 100 fps. The flight time was of approximately 19 min. Both flights were conducted in July 2020: Vallado at mid-month and Lousada in late July. As for the Viana test area—in Figure 13C—a total of 14 hyperspectral swaths were scanned, reaching a 2.5 cm GSD with a 40% sidelap, flying at a speed of 5 m/s and a height of 40 m above the ground, and scanning at a frame rate of 100 fps. The flight time was of approximately 15 min. This flight was carried out in November 2020. Figure 15 presents the three flight plans used to fly the Nano-Hyperspec sensor for both vineyards and the coastal area, prepared with the UgCS software. All the 272 spectral bands (400 nm–1000 nm, sampling interval of 2.2 nm) and a 12 mm lens with a horizontal FOV of 21.1° were used, with a FWHM of about 6 nm.

It should be noted that the scanning area must also be loaded onto the sensor, using Headwall’s Hyperspec software. This creates a geographical boundary in which the sensor turns off when leaving and on when entering, therefore avoiding the creation of frames with significant distortions.

#### 2.5.2. Senop Hyperspectral HSC-2

Piacenza—in Figure 14D—and the Chianti—in Figure 14E—flights were carried out in August 2020. Both flights had a speed of 1.8 m/s and a fly height of 32 m AGL, which generated spectral images with a GSD of approximately 2 cm. Forward overlap was of approximately 75% and side overlap was of 72%. OptoPolymer reflectance reference panels were positioned close to the take-off location. White plastic targets were positioned along the border of the study areas to establish the GCPs for the image block and in front of 60 and 27 sample vines for Piacenza and Chianti, respectively. HSC-2 was configured with 50 spectral bands (500–900) with a FWHM of about 10 nm. The integration time was 1 ms to avoid saturation effect, especially with white reference panels. The difference in acquisition time between each band was of 0.013 s, considering a frame rate of 74 fsec, and of 0.63 total sec, considering 50 bands. Figure 16 presents the flight plans used to fly the Senop Hyperspectral HSC-2 sensor over the two Italian study areas, prepared with the UgCS software.

## 3. Experimental Results

This study intends to review in detail the various stages underlying the use of each type of hyperspectral sensor, and to evaluate their overall performance regarding both operation and quality—geometric and radiometric—of the generated orthorectified data. This reason alone supports the selection of different areas in two countries, without risking any influence on both the conclusions and pertinence of this study.

Sensors’ performance comparison was carried out through the evaluation of the complexity of field operations, processing time and qualitative accuracy of the final results, namely the quality of the generated hyperspectral ortho mosaics.

Orthophoto mosaics are one of the most basic and important products generated by UAV imagery, forming also the basis of hyperspectral systems [63]. Orthorectification requires a DEM. Its quality impacts pixels geolocation on the resulting ortho mosaic. For the orthorectification of satellite-based products, the 90 m resolution SRTM DEM is normally used. However, when using high-resolution aerial imagery, such as that which is obtained by UAVs, this DEM’s quality would simple not suffice to achieve the required accuracy, especially over rough terrain. Many applications (e.g., time series analysis or data fusion with other sensors) rely on a precise georectification.

### 3.1. Nano-Hyperspec

The processing of individual campaigns/flights was performed with three different approaches: (1) no DEM; (2) using the 90 m resolution SRTM DEM; and (3) using a UAV-based high-resolution DEM (~19 cm for Vallado, ~16 cm for Lousada, and ~4 cm for Viana). Figure 17 presents the elevation profiles obtained in the slope’s direction. It is worth mentioning that in Vallado the flight lines were oriented perpendicularly to the slopes while in Lousada the flight lines were parallel.

To verify the influence of each DEM in the orthorectification process, a swath in the middle of the covered area was analyzed (the same corresponding to the profile line presented in Figure 15). Once the flight was carried out maintaining the flight height (adjusting it to the terrain), it would be expected that the orthorectification final result would originate a rectangular shape. From Table 5, it can be concluded that orthorectification performance depends on both the terrain’s average altitude and the orientation of the flight lines in relation with the slope. In fact, and as expected, when flying at sea level (Viana area), DEM’s resolution is almost irrelevant. As such, under these conditions the DEM can even be ignored. The situation is completely different in the other two study areas. The greater the slope, the greater the error associated to orthorectification process. In Vallado, which is the worst-case scenario presented with regard to land topography, DEM’s resolution plays a major role. No significant differences are detected between no DEM result and SRTM DEM. However, the results significantly improve with the use of high-resolution DEM. As for Lousada, although the altitude is in average higher than in Vallado, the use of the SRTM DEM considerably improves the orthorectification, as the slope is significantly lower. Improvements on the orthorectification can also be noted when using the high-resolution DEM, albeit less significantly when compared with Vallado results.

Table 6 presents the parameters that were used for orthorectification processing, and the estimated processing time spent to create the final mosaics. All the iterations tried that resulted in a change in the image are presented.

As part of assessing results’ accuracy, a visual evaluation was performed to detect the presence/absence of gaps, matches across boundaries, deformations, and patches. Figure 18, Figure 19 and Figure 20 present the entire orthorectified mosaics overlapped to a Google Earth (GE) image for each of the three studied areas, respectively, Vallado, Lousada, and Viana. As can be seen, generally all mosaics are free of gaps and patches, and the hyperspectral swath borders are dissembled. The zoomed areas show an overall good alignment achieved by continuous linear features, such as vineyards rows (Figure 18 and Figure 19). Shapes and sizes of individual plants and other natural and/or man-made features are well maintained, as can be seen by comparison with the GE in background.

The hyperspectral data processing was performed locally in a workstation equipped with two Intel^®^ Xeon^®^ CPU E5-2680 v4, two NVIDIA Quadro M4000 GPUs, 128 GB RAM and a 1 TB SDD in a Windows 10 Enterprise operating system.

### 3.2. Senop Hyperspectral HSC-2

Structure from Motion (SfM) process applied on snapshot sensor images enables the generation of orthorectified mosaic by matching spatial patterns of each raw image frame. Considering hyperspectral imagery acquired by UAV, the main issue is that non-aligned spectral bands are stored in each frame depending on the flight speed, movement, and vibrations of the aerial platform, which leads to a spatial shift between the single bands of the data cube as described in Section 2.3.2. The shift between bands means that the value of a pixel in the multiband mosaic does not ensure the spectral value of a given target. There are two ways to overcome this issue. The first solution is to manage each band image individually, processing groups of images relating to the same spectral band with a SfM software, and subsequently georeference all single band mosaics to obtain the final hyperspectral cube. The second approach involves a first step of co-registration of each band image in a single frame, and then reconstructing the final hyperspectral cube using multi-band images. The first solution was chosen in this work, taking advantage of the large number of GCPs distributed within both study areas to achieve an optimal georeferencing process of the single band mosaics.

For Piacenza and Chianti study areas 215 and 540 hypercube frames were processed, respectively. The pre-processing procedure described in Section 2.3.2 was carried out using Matlab, while the single band mosaic generation with Agisoft Metashape. The quality of the alignment parameter was set to “highest”, which indicated that full resolution images were used in the processing. The setting for the number of tie points per image was 4000 and a lens calibration was performed for each band. Depth maps generation parameter was set to ‘‘high” with filtering options ‘‘mild”. The total elaboration time for Piacenza and Chianti study areas are showed in Table 7, using a PC with the same specifics of Portuguese Team and with Agisoft Metashape Professional Software version 1.6.0 build 9925. A total of 50 mosaics were obtained, one per spectral band acquired. Next, the single band mosaic matching was performed by means the “georeferencing” function in QGIS using the coordinates of 60 white target panels as GCPs placed within the study areas.

The comparison presented in Table 7 enables the assessment of two very different case studies, albeit with similar spatial extension (Piacenza 1.0 ha and Chianti 1.2 ha). Considering the flight plan with the UAV’s advancing in line with the direction of the vineyard rows, it is noticeable that the sloping Chianti vineyard required a more complex route, with a high number of flight lines (29) of reduced dimensions, which resulted in a higher flight time and a larger dataset (540 frames). On the other hand, the Piacenza study area characteristics made it possible to monitor it with a lower number of flight lines (12) of larger dimensions, which led to a much smaller dataset (215 frames).

Figure 21 shows the final hyperspectral single band mosaics for the 520, 600 and 810 nm spectral bands in both Piacenza and Chianti study areas, respectively. The results show an excellent geometrical quality of the mosaicking process for each single band. Senop HSC-2 enables the reconstruction of the vineyards present in the study areas without any distortions or overlapping errors.

The next step is the construction of the multilayer hyperspectral mosaic. An accurate overlapping process of each mosaic must be carried out in what is considered critical in the processing workflow of Senop HSC-2 hypercubes generation. Figure 22 shows a detail of the three-band mosaic obtained by merging the unmatched mosaics related to the three previously identified band (520, 600 and 810 nm) in both Piacenza (Figure 22a) and Chianti (Figure 22c). This process is resolved through a georeferencing operation that allows the creation of a co-registered multilayer mosaic. The good quality of the results is visually highlighted both in Piacenza (Figure 22b) and Chianti (Figure 22d) by observing the linear reconstruction of the vineyard rows and the matching of GCPs and panels as indicators.

## 4. Qualitative Analysis

This work begun to be thought of by two research groups—a Portuguese and an Italian—when discussing each group’s workflow to both acquisition and processing of hyperspectral UAV-based imagery, as well as results obtained in precision viticulture applications. Both groups were fully equipped to acquire hyperspectral imagery with a UAV-coupled sensor. While the UAV of choice was the same—DJI Matrice 600 Pro—, the hyperspectral sensors were not: the Portuguese group is equipped with Headwall’s Nano-Hyperspec and the Italian group with Senop’s Hyperspectral HSC-2. The former is a push-broom and the latter a single-shot type sensor. The type of sensor greatly affects field operations, processing performance and the quality of the final product. Learning the hyperspectral data acquisition process for both approaches, while highlighting their differences and comparing results, is a contribution for both researchers and other stakeholders when selecting the most suitable sensor for their own purposes.

Albeit a comparative study should unquestionably reduce the number of different variables and parameters to preferably none other than the elements to be compared—in this specific experiment, it means using the same UAV over the same study areas, while trying to consistently have equal context variables, namely environmental and lighting conditions to compare uniquely both sensors—the fact is that the research groups are from different countries, which would amount to a (passable) logistic issue. Despite not the same study areas, each sensor operation and output’s geometric and radiometric quality can be presented and assessed in very similar conditions (Table 4). To evaluate terrains’ slope influence in each sensors’ operation, both a flat—Lousada, Portugal and Piacenza, Italy—and more rugged—Vallado, Portugal and Chianti, Italy—study areas were selected in each country. In Portugal, another study area was added—Viana do Castelo, a coastal terrain—to properly assess the rather essential role that DEMs play in the orthorectification process of Nano-Hyperspec swaths.

While the Italian group has an RGB camera also coupled to the UAV (in addition to the Senop Hyperspectral HSC-2), which allows them to acquire both DEM required data and the hyperspectral imagery in the same flight, the Portuguese group had to perform two flights in the studied areas. However, this is merely a set up issue, as it does not bare any influence in the outcome. Indeed, each group obtains separate data for the DEM and for the hyperspectral cube. To verify that both relief and altitude have no effect an ASL area—Viana do Castelo—was also studied.

As for radiometric calibration, there are some differences between the process required for each sensor. While both require the assessment of dark current—which is carried out by covering the optics with a cap prior to each flight—white reference and spectra correction are achieved otherwise: Headwall’s Nano-Hyperspec optical system requires a white target and a three different gray reflection levels tarp placed on the studied area, while Senop Hyperspectral HSC-2 uses white panels—60 in Piacenza and 27 in Chianti—together with (three) reflectance panels, all placed within the studied area. It is true that the number of white panels can be reduced to between 5 and 13 if properly placed. Still, it is indisputable that the logistics involved in having a Senop Hyperspectral HSC-2 sensor acquiring hyperspectral imagery over an area is more challenging than with Headwall’s Nano-Hyperspec.

Regarding the data acquisition processes, given that the UAVs used by both groups are the same, flight plans depend only on a few other parameters, such as the study area dimension and orography, as well as the required detail level and images’ overlap. Considering that the flight plan for each studied area was designed to obtain quality and precise outputs, they should not be factored in when pondering about each sensor’s performance. However, one important aspect to be considered is the number of bands retrieved in each flight. Indeed, whereas Senop Hyperspectral HSC-2 acquires 50 chosen bands, Headwall’s Nano-Hyperspec acquires 272 bands. Still addressing performance, Senop Hyperspectral HSC-2 showed significantly higher processing times than those obtained with the Headwall’s Nano-Hyperspec when generating orthorectified mosaics. As an example, Vallado and Chianti have many similarities: both sloping vineyards monitored with similar forward speed (2 m/s and 1.8 m/s, respectively), flight height of about 32 m and ground resolution of about 2 cm/pixel. Under these similar conditions, Vallado data required approximately 8 h of processing for a 272-band mosaic (0.029 h/band), while Chianti required approximately 12.5 h for a 50 spectral bands mosaic (around 0.25 h/band).

Regarding the visual quality of the outputs, Senop HSC-2 enables the reconstruction of the vineyards present in the study areas without any distortions or overlapping errors, while Headwall’s Nano-Hyperspec presented some distortions in the Vallado study area. This is the worst-case scenario presented regarding land topography, and for that reason DEM’s resolution plays a major role in the orthorectification performance. No significant differences are detected between no DEM result and SRTM DEM. However, the results significantly improve with the use of high-resolution DEM. As for Lousada, although the altitude is in average higher than in Vallado, the use of the SRTM DEM considerably improves the orthorectification, as the slope is significantly lower. Improvements on the orthorectification can also be noted when using the high-resolution DEM, albeit less significantly when compared with Vallado results. It is thus possible to conclude that although the absolute altitude at which the flight is carried out has implications in the orthorectification process, topographic variability has an even more significant effect. Therefore, for most applications it is recommended to use the UAV-based high-resolution DEM to take advantage of push-broom hyperspectral high spectral and spatial resolutions. In these cases, the presented methodology showed that higher-resolution DEMs improve the final orthorectification output. By analyzing Table 7, it is possible to conclude that the processing complexity increases with the characteristics of the terrain. The more rugged topography, the more adjustments/iterations will be necessary. Furthermore, the various parameters are distinct in various swaths. Obviously, this has an impact on the final processing time, which in this case ranges from 2 h (Viana area) to 8 h (Vallado area).

However, topography remains a determining factor and it is possible to detect some cases, especially in the Vallado area (Figure 18), where some difficulties have not been fully resolved. These problematic areas are more noticeable at the mosaic’s borders, especially when frames were incomplete, which makes adjustments more challenging. The Viana study area (Figure 20), flat and at sea level, presents the best fit. In this case, it is not possible to detect any discontinuities, even at the borders of the mosaic. Whereas the single band mosaicking procedure (Senop HSC-2) provides excellent results, the main issue is the time required to create each mosaic from the single bands images. The quality of the final hypercube mosaic does also need a correct georeferencing process of the individual mosaics, which requires the presence of GCPs distributed within the study areas. Excellent results were obtained using 60 targets in the Piacenza study area. However, an accurate overlap was obtained even using a lower number of GCPs, as demonstrated by the Chianti case study, in which 27 targets were used. Considering this aspect, Harwin et al. [64] reported that between 5–13 GCPs are sufficient, if distributed along the outside of the study area, with at least one GCP in its center.

Push-broom and snapshot approaches for acquiring spectral imagery has both pros and cons (Table 8). In theory, no approach is better. As such, the one to be used should be selected depending on the application and defined goals. Push-broom uses a dispersive element (prism, grating, prism-grating-prism–PGP) to divide the light, which offers an advantage of uniformly high efficiency, low dispersion, and low cost for this type of imager. Another advantage is that acquired images reflect only the y–λ plane, enabling a single line of light as the image’s source of illumination [27]. This approach is especially suitable for applications involving the search for objects with specific spectral signatures. Indeed, the entire spectrum of each pixel in the image of the same range is available in real time. Nonetheless, push-broom also has disadvantages. The main one is the optical design complexity. Moreover, the time required to acquire a data cube is long, when compared to conventional mechanical scanning systems. In practice, this last disadvantage is reflected in the speed of the platform that carries the sensor, depending on the lighting conditions.

As for snapshot, the fact that it does not require a scanning system in any of the domains—spatial or spectral—to obtain a spectral cube makes it very attractive in applications that require fast spectral image acquisitions. Therefore, it is an ideal solution for applications where imaging time is an important factor. As a limitation, it can be highlighted that these systems are still in development, with restraints regarding scope, spatial and spectral resolutions [57]. The system design is generally more complex than that for scanning systems and uses recent technology, such as large focal plane array (FPAs), high-speed data transmission, advanced manufacturing methods, and precision optics. Only recently commercial data transmission formats have become fast enough to fully utilize a large-format snapshot imaging spectrometer for daylight scenes. Thus, snapshot’s advantage when compared to push-broom is the efficiency of using incident light: push-broom uses only a small fraction of the incident light to generate the image. There are other obvious advantages of snapshot: (i) image generation is less complex; (ii) the three-dimensional data [x, y, λ] is obtained with a single reading of the sensor, with no need for any type of image combination to generate the hyperspectral cube; (iii) less influence of moving artifacts and all channels and locations are captured at the same time; (iv) due to the capture speed and by definition, there can be no moving parts, making the whole configuration more robust and less complex.

## 5. Conclusions

UAV-acquired hyperspectral imagery is slowly gaining momentum as the basis to attain data that is otherwise unavailable by using any other available sensor. Within precision agriculture, the focus has been on detecting plagues and diseases before symptoms are visually detectable on plants, which would enable timely and more efficient treatments. It is still a research area were much work needs to be carried out and that is now becoming increasingly accessible mostly due to the decrease in hyperspectral sensors’ cost. However, the state-of-the-art seems to have a fundamental lack of works that may be of help to other researchers and to the industry by specifying important features, advantages and disadvantages of available hyperspectral sensors, when dealing with precision agriculture research. In this work we present the process upon which mosaics were obtained for five study areas, using two different hyperspectral sensors also representing two different approaches for data acquisition—push-broom and a snapshot—in four areas—two for each sensor—that shared similar geographic characteristics, the same crop and crop-structure. The remainder area—coastal—enabled the demonstration of DEM’s influence on push-broom swaths’ orthorectification process. The aim of the presented study was not to provide a comparison between types of hyperspectral sensors or even models but provide researchers with a reference guide for applications that need to resort to hyperspectral data. Indeed, learning fully the process that leads to the generation of the hyperspectral mosaic, depending on the sensor’s data acquisition approach, is of utmost importance. In fact, this allows access to the main advantages and disadvantages of each sensor, from the point of view of the functioning of the sensor itself and the characteristics of the data, but also allows to understand the operational complexity depending on the precision agriculture context. In this way, the reader and/or the future owner of a UAV-based hyperspectral sensor will be aware of things to watch for or to correct to have quality data. Albeit, both the performance and singularities of each sensor were thoroughly analyzed in the previous section, some essential bits are worth to mention. The push-broom sensor can acquire its full spectral range within a single flight and requires less for processing time to obtain the mosaic when compared with the snapshot sensor. This conclusion was reached using similar geographic characteristics, the same crop and crop-structure. However, push-broom often presents misalignments on the edge of the mosaics. This may be surpassed by planning flights that cover areas slightly larger than the area of interest and then crop the mosaic without potential edge misalignments. As for the snapshot sensor, it can acquire a previously selected spectral range in a single point in the flight (not the full available range). Notwithstanding rendering quality outputs, it requires a hefty logistical operation in the field by placing ground references. While both sensors are able to do their job, this technical note may be proven valuable to help guide readers in choosing the more suitable for their objectives, while considering their own requirements.

## Figures and Tables

**Figure 1 sensors-22-06574-f001:**
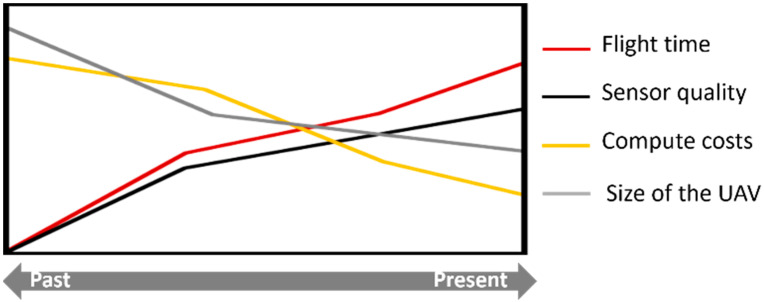
The evolution of UAV-based remote sensing in terms of associated costs, sensors’ quality and UAVs’ autonomy (adapted from PrecisionHawk [17]).

**Figure 2 sensors-22-06574-f002:**
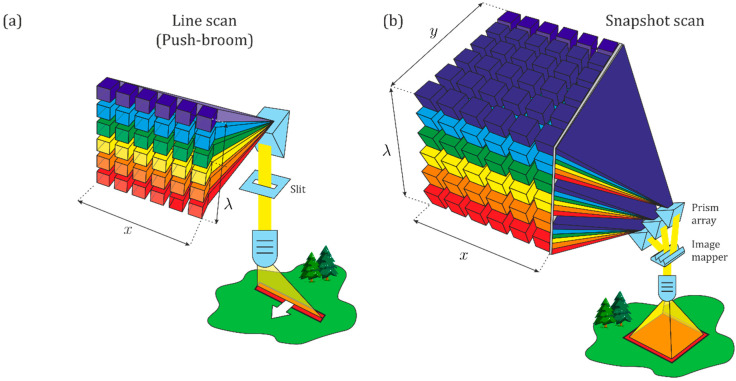
Portions of the data cube collected during a single detector integration period for (**a**) push-broom: spatial dimension (y-line), spectral dimension (λ) and (**b**) snapshot devices: spatial dimension (x, y), spectral dimension (λ) (adapted from Jurado et al. [25]).

**Figure 3 sensors-22-06574-f003:**
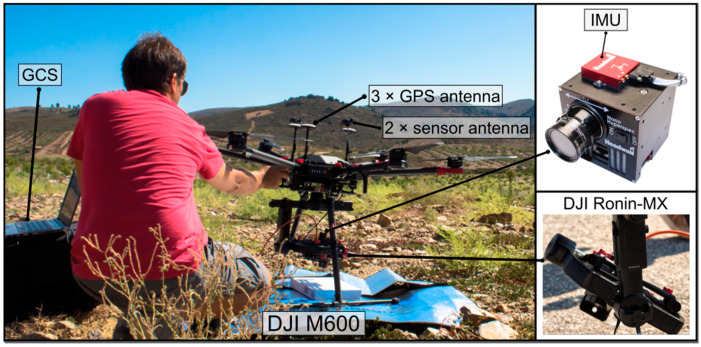
Setup of the DJI Matrice 600 UAV system and Nano-Hyperspectral sensor. The computer on the figure’s left side is used as ground control station (GCS), allowing to control all mission’s operations.

**Figure 4 sensors-22-06574-f004:**
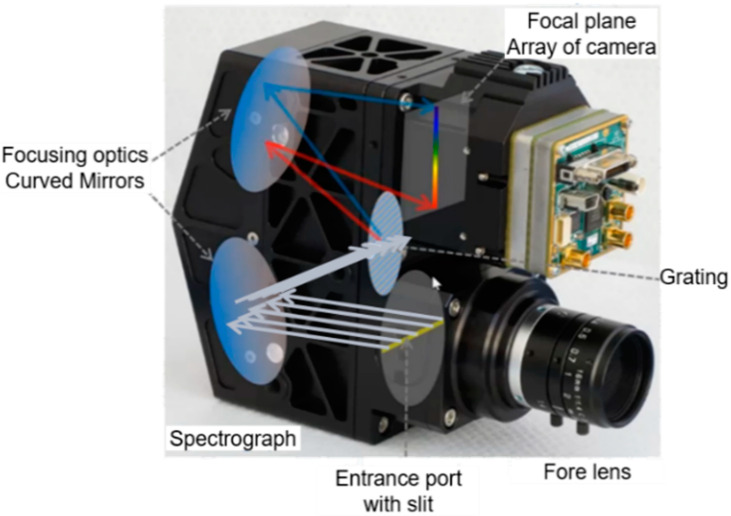
The optic system inside the Nano-Hyperspectral sensor based on offner spectrometer: the light passing through the slit is projected in the focal plane (camera array of CCD) with almost no distortions. Next, the light is separated according to the wavelength creating a hyper pixel. Adapted from Headwall Photonics, Inc.

**Figure 5 sensors-22-06574-f005:**
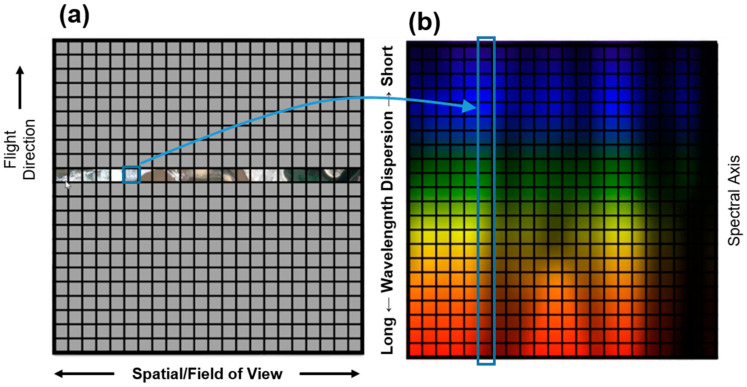
Operation principle (viewed through the slit): (**a**) spatial information collected, corresponding to the area scanned by the sensor—light passing the slit; (**b**) light passing the slit is dispersed on 272 bands, from the short to the long wavelength. Adapted from Headwall Photonics, Inc.

**Figure 6 sensors-22-06574-f006:**
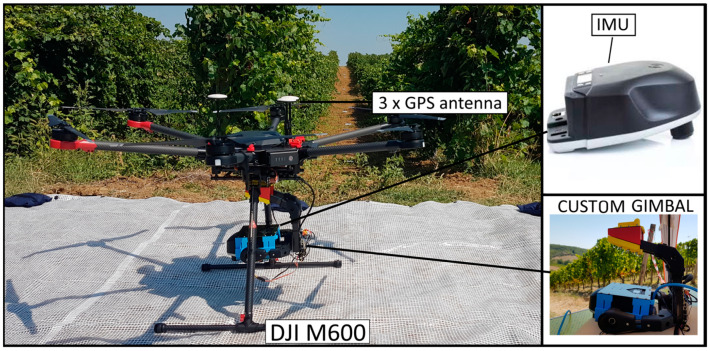
Setup of the DJI Matrice 600 UAV system and Hyperspectral HSC-2 sensor mounted on a self-developed gimbal.

**Figure 7 sensors-22-06574-f007:**
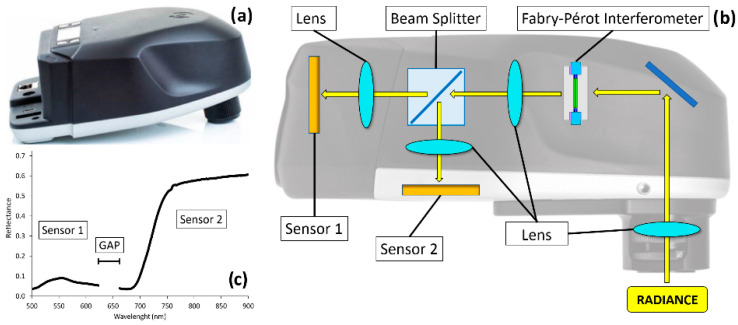
Senop Hyperspectral sensor HSC-2 (**a**) with details of the optic system based on Fabry-Pérot Interferometer (FPI) technology, a beam splitter and two CMOS sensors (**b**). The graph (**c**) shows a reflectance vegetation signature composed by the two sensors, and the “gap-area” placed between them (@ 636 nm) with invalid data.

**Figure 8 sensors-22-06574-f008:**
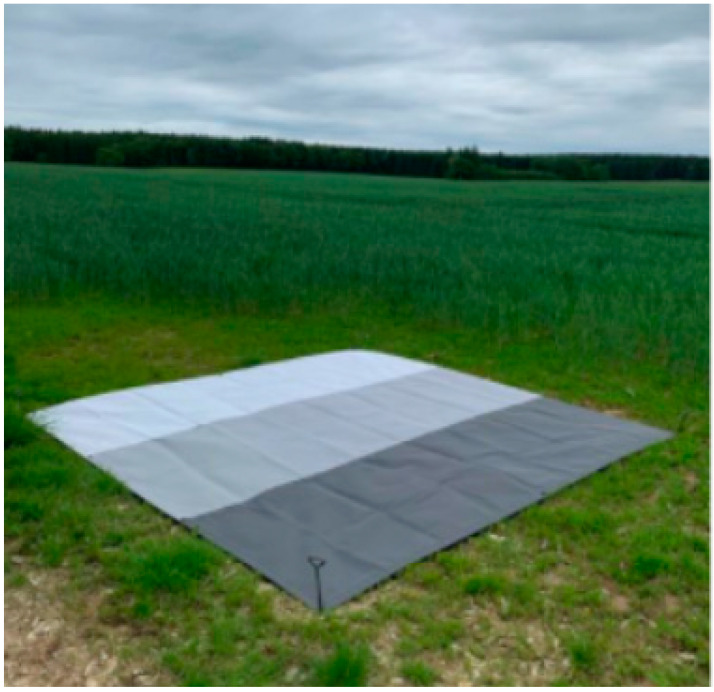
Tarp with the three gray reflection levels deployed at the interest area, used for radiometric calibration.

**Figure 9 sensors-22-06574-f009:**
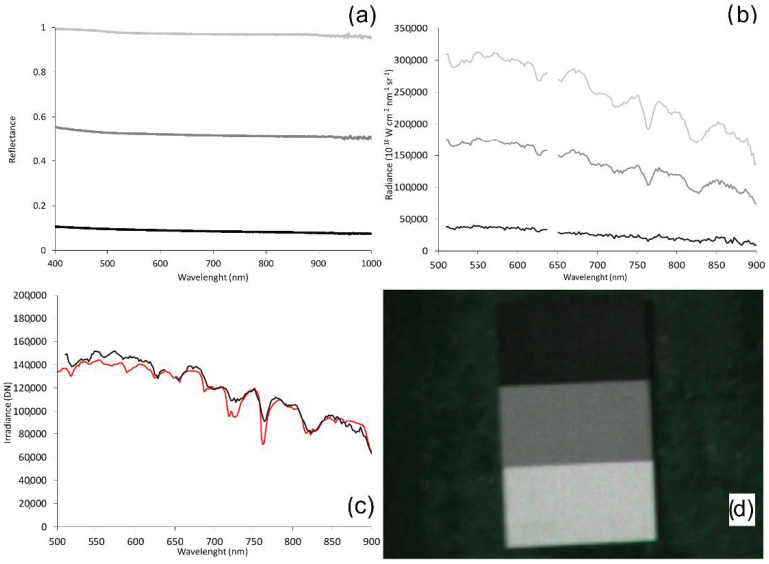
(**a**) Reflectance spectral signature from 400 nm to 1000 nm of the control targets (OptoPolymer panels) obtained in the field with the spectroradiometer (GER); (**b**) Radiance spectral signature of the control targets (OptoPolymer panels) obtained in the field with Senop HSC-2; (**c**) Irradiance spectral signature (DN) comparison between GER spectroradiometer (red line) and Senop HSC-2 (black line); (**d**) OptoPolymer reference panels in the field. Reference panels legend: white panel @ 97% reflectance (light grey line), grey panel 56% reflectance (dark grey line), black panel @ 10% reflectance (black line).

**Figure 10 sensors-22-06574-f010:**
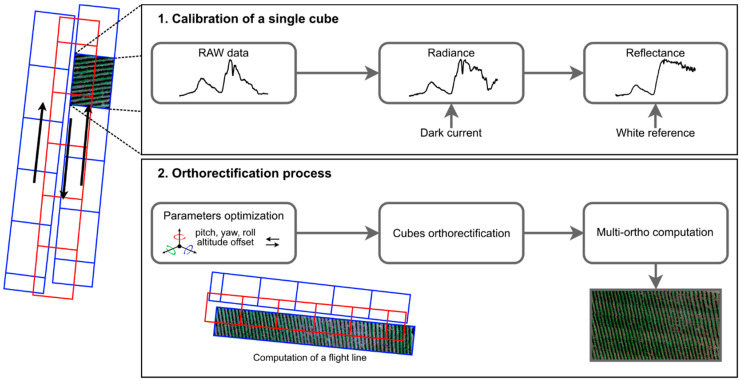
Nano-Hyperspec processing workflow.

**Figure 11 sensors-22-06574-f011:**
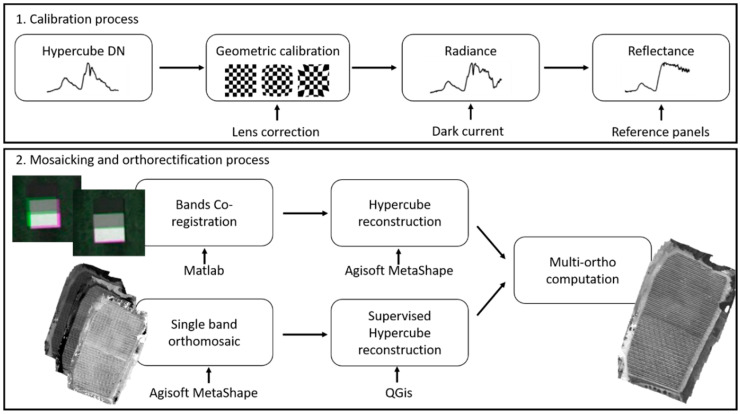
Senop HSC-2 processing workflow.

**Figure 12 sensors-22-06574-f012:**
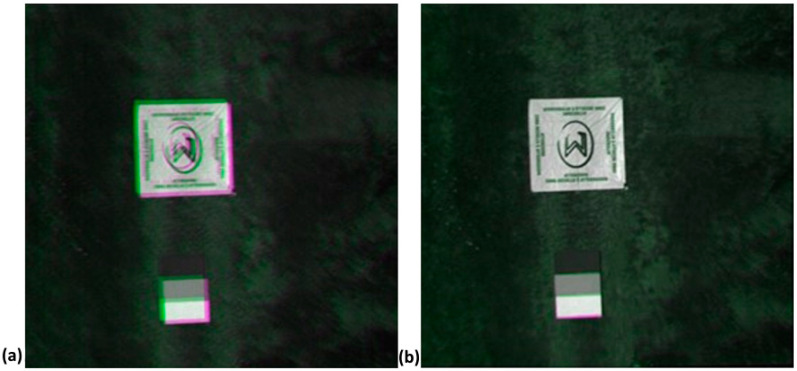
Example of co-registration images result: (**a**) no co-registrated bands and (**b**) co-registrated bands.

**Figure 13 sensors-22-06574-f013:**
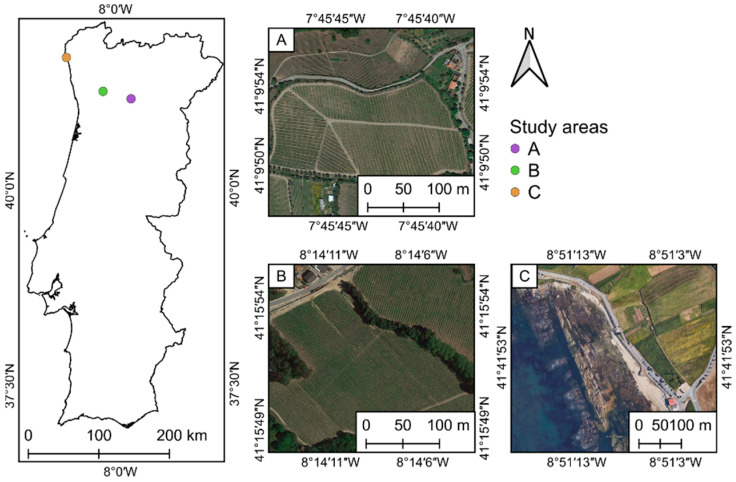
Location of the Portuguese study areas: (**A**) 1.5 ha vineyard parcel, belonging to Quinta do Vallado, Peso da Régua, Portugal; (**B**) 2 ha vineyard parcel, in Lousada, Porto, Portugal; and (**C**) 5 ha coastal area, located in Viana do Castelo, Portugal. Images from Google Earth (accessed on 18 December 2020).

**Figure 14 sensors-22-06574-f014:**
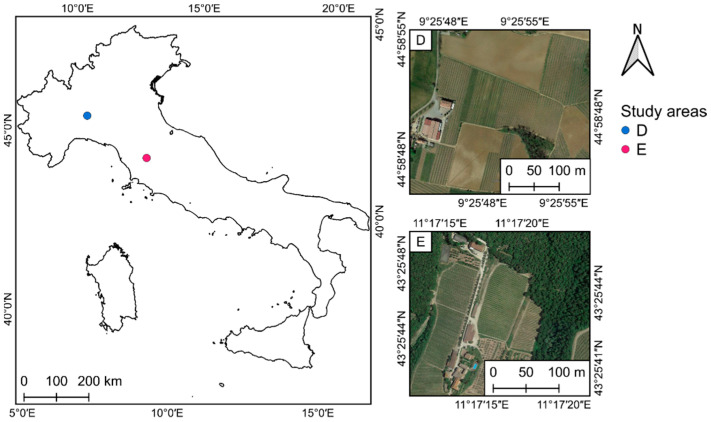
Location of the Italian study areas: (**D**) 1 ha Piacenza vineyard parcel—215 m ASL—belonging to Tenuta La Pernice, Borgonovo Val Tidone, Piacenza, Italy; and (**E**) 1.2 ha Chianti vineyard parcel—355 m ASL—in the Castello di Fonterutoli farm, Castellina in Chianti, Siena, Italy. Images from Google Earth (accessed on 18 December 2020).

**Figure 15 sensors-22-06574-f015:**
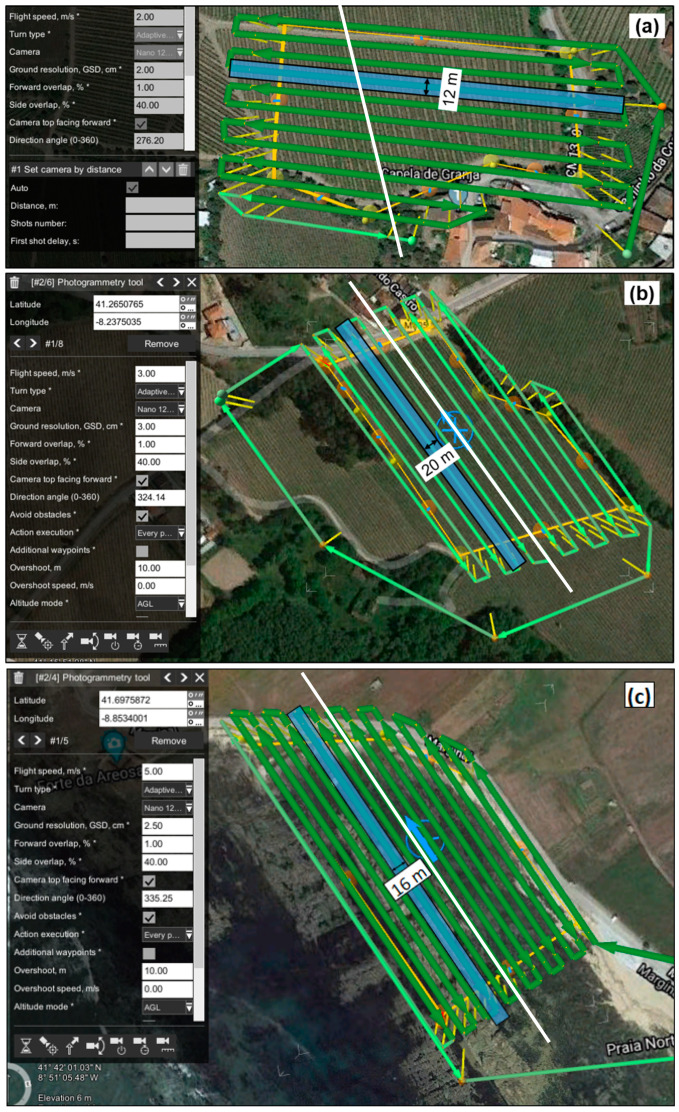
Flight plans, parameters and mission area for the Portuguese study areas: (**a**) Vallado; (**b**) Lousada; and (**c**) Viana. The white dashed lines represent the elevation profile lines.

**Figure 16 sensors-22-06574-f016:**
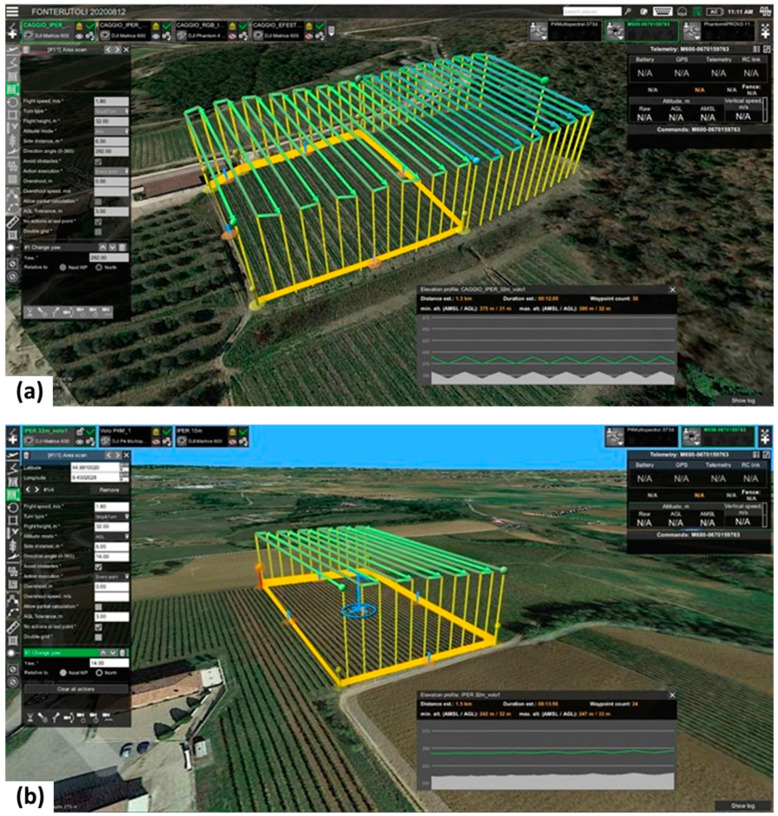
Flight plans, parameters and mission area for the Italian study areas: (**a**) Piacenza test area; and (**b**) Chianti test area.

**Figure 17 sensors-22-06574-f017:**
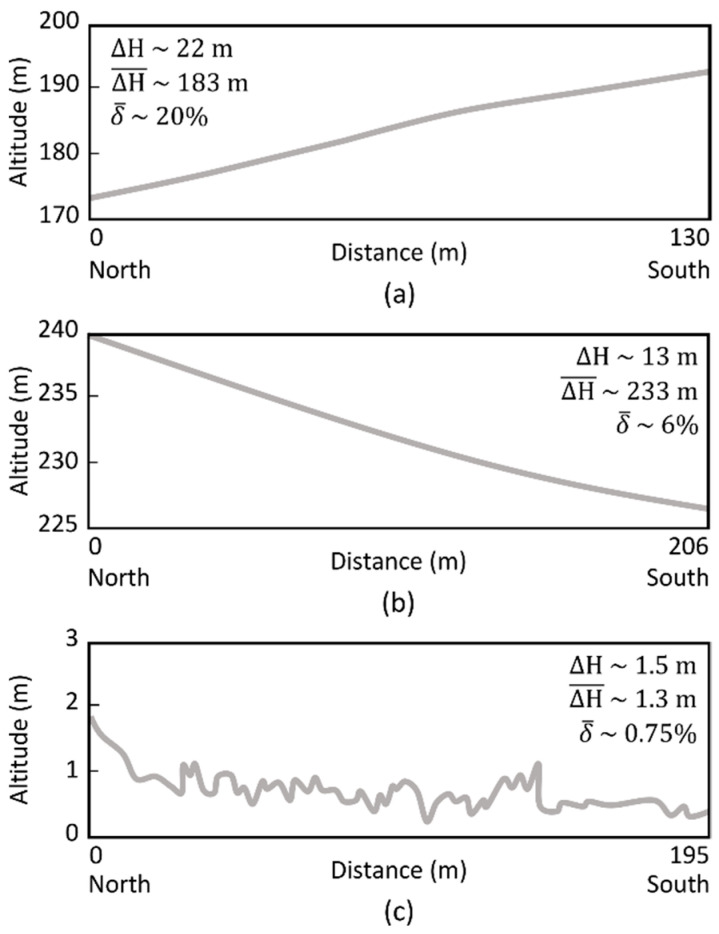
Elevation profile lines of the scanned areas: (**a**) Vallado; (**b**) Lousada and; (**c**) Viana. ΔH represents the altitude difference, $\bar{\Delta H}$ represents the terrain average altitude, and $\bar{\delta }$ represents the terrain average slope.

**Figure 18 sensors-22-06574-f018:**
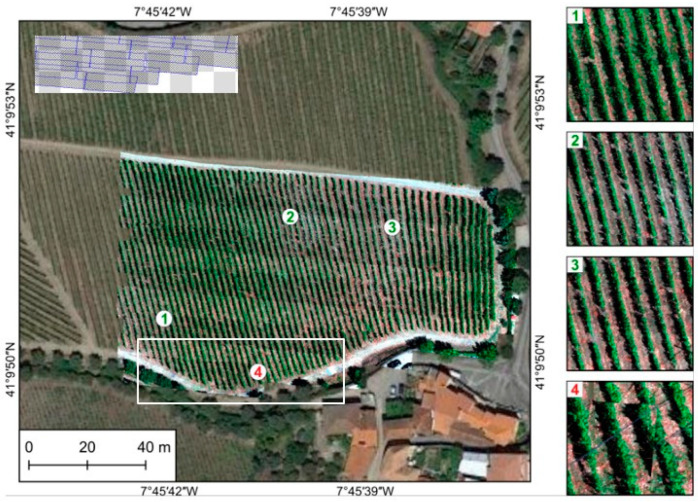
Vallado area: overview of the final hyperspectral mosaic and close-ups to show the good alignment achieved by continuous linear features (**1**–**3**). Few misalignments are still present, but only at the edge of the mosaic (**4**). In the upper left corner, hyperspectral frames coverage from the white rectangle. Incomplete frames make its adjustment more complex, as in (**4**).

**Figure 19 sensors-22-06574-f019:**
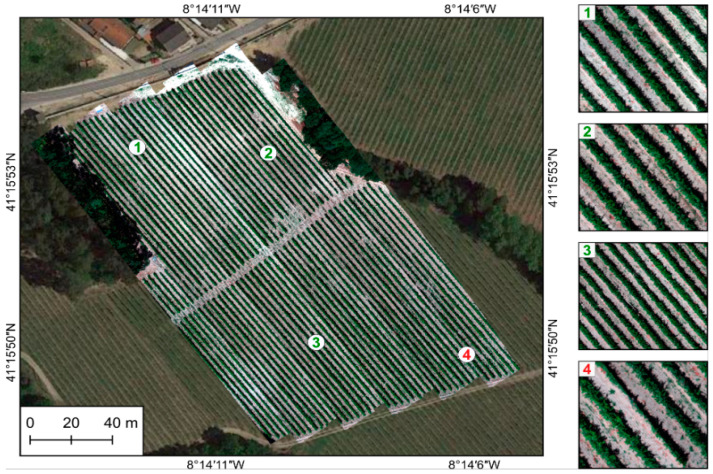
Lousada area: overview of the final hyperspectral mosaic and close-ups to show the good alignment achieved by continuous linear features (**1**–**3**). Few misalignments are still present, but only at the edge of the mosaic (**4**).

**Figure 20 sensors-22-06574-f020:**
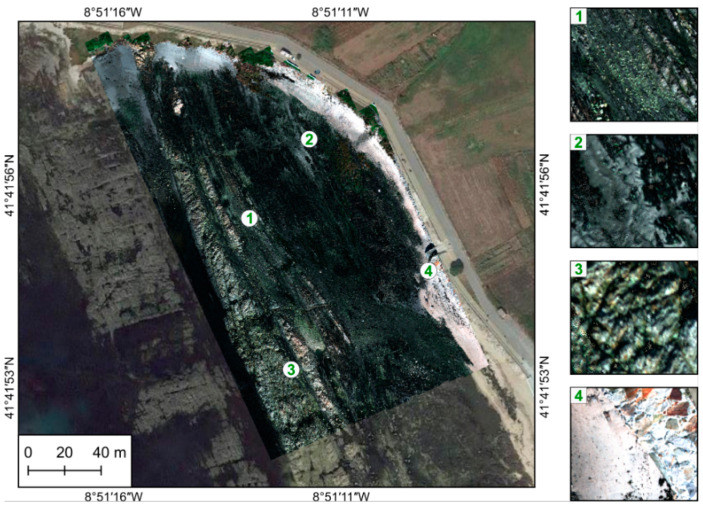
Viana area: overview of the final hyperspectral mosaic and close-ups to show the good alignment achieved (**1**–**4**). No misalignments are visible.

**Figure 21 sensors-22-06574-f021:**
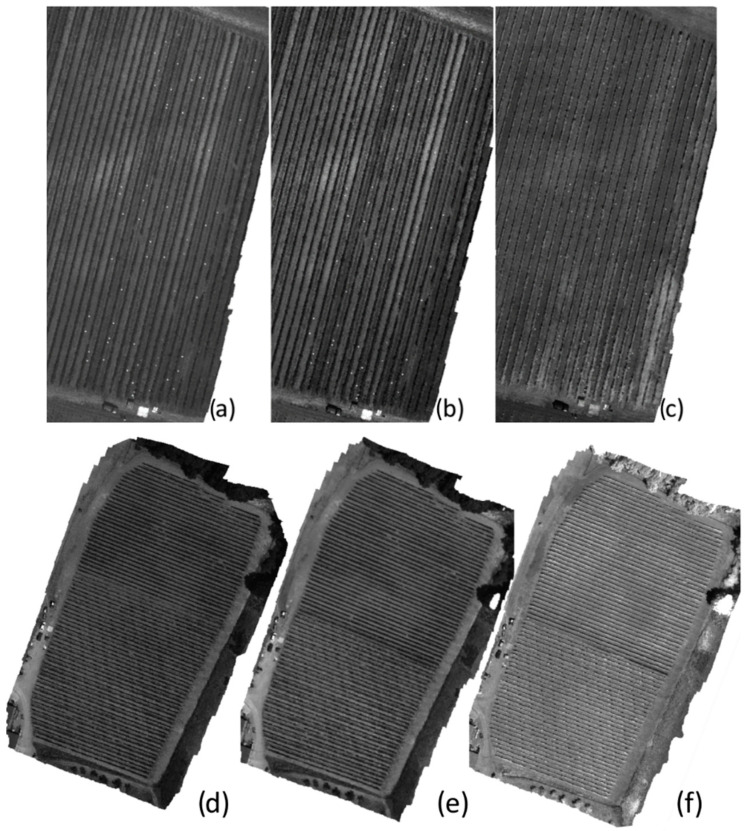
Overview of the single band mosaics (**a**–**d**) band 520 nm; (**b**–**e**) band 600 nm and (**c**–**f**) band 810 nm, for both Piacenza and Chianti study areas, respectively. Map details related to scale and geographic coordinates are reported in Figure 14.

**Figure 22 sensors-22-06574-f022:**
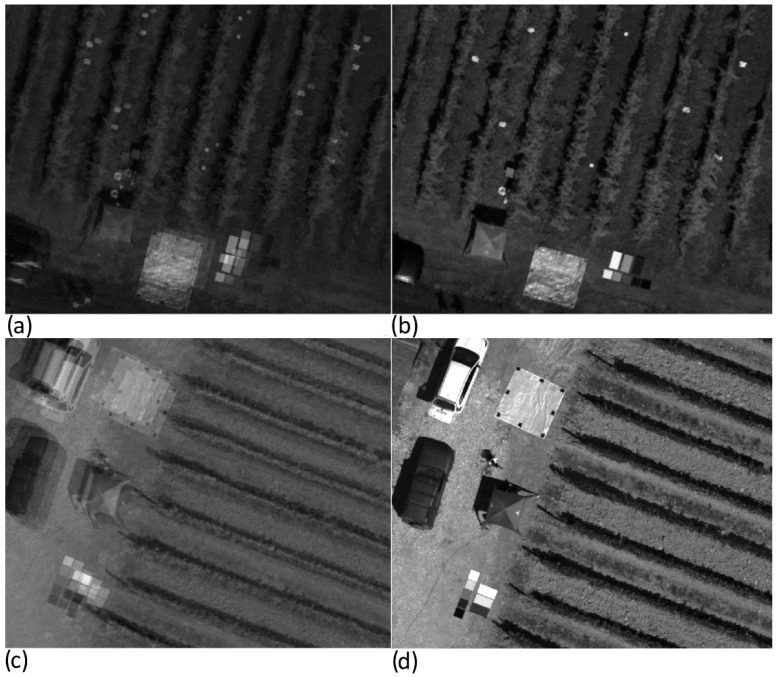
Details of three bands composite results for Piancenza (**a**,**b**) and Chianti (**c**,**d**) study areas: (**a**,**c**) unmatched band composite using the original mosaics showed in Figure 21; (**b**,**d**) using the three bands’ mosaics after the georeferencing procedure in QGIS software.

**Table 1 sensors-22-06574-t001:** Characteristics of the main commercially available hyperspectral sensors developed (or with potential) to be coupled with UAVs (adapted from Adão et al. [24]). The two sensors used in this study are highlighted in bold.

Manufacturer	Sensor	Spectral Range (nm)	No. of Bands	Spectral Resolution (nm)	Spatial Resolution (px)	Acquisition Mode	Weight (kg)
BaySpec	OCI-UAV-1000	—	100	<5	2048	Push-broom	0.272
Brandywine Photonics	CHAI S-640	—	260	2.5–5	640 × 512	Push-broom	5
	CHAI V-640	—	256	5–10	640 × 512	Push-broom	0.48
Cubert GmbH	S 185—FIREFLEYE SE	450–950 355–750	125	4	50 × 50	Snapshot	0.49
	S 485—FIREFLEYE XL	450–950 550–1000	125	4.5	70 × 70	Snapshot	1.2
	Q 285—FIREFLEYE QE	450–950	125	4	50 × 50	Snapshot	3
**Headwall** **Photonics Inc.**	**Nano HyperSpec**	**400–1000**	**272–775**	**6**	**640**	**Push-broom**	**1.2**
	Micro Hyperspec VNIR	380-1000	837–923	2.5	1004 1600	Push-broom	~3.9
HySpex	VNIR-1024	400–1000	108	5.4	1024	Push-broom	4
	Mjolnir V-1240	400–1000	200	3	1240	Push-broom	4.2
	HySpex SWIR-384	1000–2500	288	5.45	384	Push-broom	5.7
NovaSol	vis-NIR microHSI	400–800	120	3.3	680	Push-broom	~0.45
		400–1000	180				
		380–880	150				
	Alpha-vis micro HSI	400–800	40	10	1280	Push-broom	~2.1
		350–1000	60				
	SWIR 640 microHSI	850–1700	170	5	640	Push-broom	3.5
		600–1700	200				
	Alpha-SWIR microHSI	900–1700	160	5	640	Push-broom	1.2
	Extra-SWIR microHSI	964–2500	256	6	320	Push-broom	2.6
PhotonFocus	MV1-D2048x1088-HS05-96-G2	470–900	150	10–12	2048 × 1088	Push-broom	0.265
Quest Innovations	Hyperea 660 C1	400–1000	660	-	1024	Push-broom	1.44
Resonon	Pika L	400–1000	281	2.1	900	Push-broom	0.6
	Pika CX2	400–1000	447	1.3	1600	Push-broom	2.2
	Pika NIR	900–1700	164	4.9	320	Push-broom	2.7
	Pika NUV	350–800	196	2.3	1600	Push-broom	2.1
**SENOP**	**VIS-VNIR Snapshot**	**400–900**	**380**	**10**	**1010 × 1010**	**Snapshot**	**0.72**
SPECIM	SPECIM FX10	400–1000	224	5.5	1024	Push-broom	1.26
	SPECIM FX17	900–1700	224	8	640	Push-broom	1.7
Surface Optics Corp.	SOC710-GX	400–1000	120	4.2	640	Push-broom	1.25
XIMEA	MQ022HG-IM-LS100-NIR	600–975	100+	4	2048 × 8	Push-broom	0.032
	MQ022HG-IM-LS150-VISNIR	470–900	150+	3	2048 × 5	Push-broom	0.3

**Table 2 sensors-22-06574-t002:** Nano-Hyperspec sensor specifications.

**Wavelength Range (nm)**	400–1000	**Camera Technology**	CMOS
**Spatial bands**	640	**Bit depth**	12-bit
**Spectral bands**	272	**Max Frame Rate (Hz)**	350
**Dispersion/Pixel (nm/pixel)**	2.2	**Detector pixel pitch (µm)**	7.4
**FWHM Slit Image (nm)**	6	**Max Power (W)**	13
**f/#**	2.5	**Storage capacity (GB)**	480
**Entrance Slit width (µm)**	20	**Operating Temperature (ºC)**	0–50
**Weight without lens (kg)**	0.5	**GPS**	Integrated

**Table 3 sensors-22-06574-t003:** Senop hyperspectral HSC-2 sensor specifications.

**Optics**	F#3.28	**Image Frame Size (Pixels)**	1024 × 1024
**FOV**	36.8°	**Frame rate** **(single image or video)**	12-bit frame: max 74 f/s 10-bit image: max 149 f/s
**Focus distance**	30 cm to ∞	**Wavelength area (nm)**	500–900 (up to 1000 spectral bands)
**Exposure time**	can be set freely	**Spectral FWHM bandwidth**	Narrow < 15 nm, Normal < 20 nm, Wide < 25 nm
**Integrated IMU**	Accuracy typically 1°	**Storage**	1TB (Internal Hard Drive)
**Size (mm)**	199.5 × 130.9 × 97.2	**GPS**	Integrated GPS
**Weight (g)**	990	**Antenna**	Internal antenna for GPS

**Table 4 sensors-22-06574-t004:** Comparison between study areas’ structure, dimension, and orography. Both the Portuguese and Italian study areas have a flat and a more rugged terrain, with comparable dimensions and vineyard structure.

Study Area	Area (ha)	Slope (%)	Spacing (m) Inter-Row × Intra-Row
Vallado	1.5	20	2.2 × 1.3
Chianti	1.2	17	2.2 × 0.75
Lousada	2	flat	2.8 × 1.75
Piacenza	1	flat	2.4 × 1.0

**Table 5 sensors-22-06574-t005:** Swath fit after orthorectification, in each study site, when using different digital elevation models (DEMs) or no DEM.

Area	No DEM	SRTM DEM	UAV DEM
Vallado	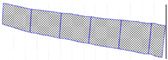	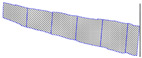	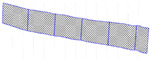
Lousada	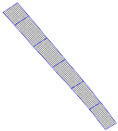	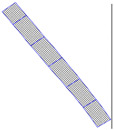	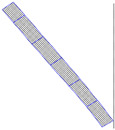
Viana	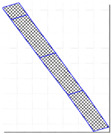	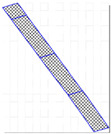	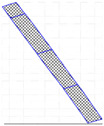

**Table 6 sensors-22-06574-t006:** Parameters used and total processing time of the final hyperspectral ortho mosaics. A single value in a cell means the same value has always been used. The altitude offset is set up subtracting the geoid undulation to the average altitude of the area.

Study Area	Parameters				Processing Time (Hours)	Mosaic Size (Gbyte)
	Pitch (Deg)	Roll (Deg)	Yaw (Deg)	Alt. Offset (m)		
Vallado	0	0	0	−125	8	~60
	1.8	0	−2			
	0	0	0			
	−3	4	0			
	−0.5	0	2			
	0	−2	0			
	1	0	−1			
	1	2	0			
	0	0	1			
	1	−1	−1			
Lousada	−1	−1	0	−170	7	~65
	0	−2	0			
	−1	−2.3	0			
	0	−2	0			
	−1.2	−1.5	0			
	0	−2	0			
	−1.2	−1.5	0			
	0	−2	0			
	−1.2	−1.5	0			
	0	−2	0			
Viana	0	0	0	−60	2	~80

**Table 7 sensors-22-06574-t007:** Dataset characteristics and processing time of the final hyperspectral ortho mosaics.

Study Area	Pixel Res. (cm)	Frame Size of Each Band	No. of Frames	No. of Bands	Mosaic Size (Gbyte)	Pre-Processing Time (h)	Mosaic Processing Time per Band
Piacenza	2	8 MB	215	50	8.5	2	12 min
Chianti	2	8 MB	540	50	11.5	2.5	15 min

**Table 8 sensors-22-06574-t008:** Comparison between push-broom and snapshot approaches, regarding their advantages and limitations. For a given evaluated parameter, + indicates lower/medium performance while +++ indicates high performance.

**Sensor Type**	Push-Broom	Snapshot
**Acquisition design**	Scanner	Optical
**No. of bands**	++	+++ *
**Spectral coverage**	++	++
**Processing time**	+++	++
**Bands co-registration**	+++	+
**Processing complexity**	+ (multiple swaths) +++ (single swath)	+
**Flight planning**	+++	++
**Limitations**	-Spatial distortions -Data alignment with other sensors -Orthorectification in topographically complex areas	-Radiometric calibration -Bands co-registration -GCPs dependent -Complex field operation
**Advantages**	Bands co-registration	Simple individual bands processing (compared to RGB)

* The image can be formed from up to 1000 freely selectable spectral bands with 0.1 nm precision, however, handling complexity and processing time increase in proportion to the number of bands used. In this study, only 50 bands were used.

## Data Availability

The data presented in this study are available on request from the corresponding author.

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
