# Peer review of "UAV-Based Hyperspectral Monitoring Using Push-Broom and Snapshot Sensors: A Multisite Assessment for Precision Viticulture Applications"

_sensors, 2022, doi:10.3390/s22176574_

Round 1

Reviewer 1 Report (New Reviewer)

This paper offered good info on UAV hyperspectrum payloads. Table-1 is informative. The assessment steps are logically presented with real field data.

I think this paper is publishable in Sensors. Table 8 serves as a guideline for sensor comparisons.

Overall, this paper is well organized and contains useful info for readers of Sensors.

Author Response

Reviewer 2 Report (New Reviewer)

line 326. Add necessary references for this process. I recommend some ones:

This manuscript tried to compare two different hyper-spectral sensors. Nowdays, the UAV-based multis-spectral and hyper-spectral sensors were widely applied. This paper paid too much attention to the detail of the preliminary work before data collection, and they have ingored the real different of the spatial and spectral different between the two different hyper-spectral sensors. Many of the sentences were common-sense to the user of UAV remote sensing, and there is no need to describe them in detail. This paper is more like a technological report rather than a scientific report.  I personally the rejection of this manuscript.

Author Response

Reviewer 3 Report (New Reviewer)

The authors examine in depth those novel technologies that aim to provide researchers, agronomists, winegrowers and UAV pilots with dependable data collection protocols and methods to achieve faster processing techniques and integrate multiple data sources in precision viticulture remote sensing applications.

The results look encouraging and motivating. But there are still some contents, which need be revised in order to meet the requirements of publish. A number of concerns listed as follows:

(1)   The abstract should be rewritten to reflect the significance of the proposed work.

(2)   In the introduction, it is necessary to add a research background introduction and a detailed explanation of the research motivation, so as to attract more potential audiences.

(3)   The methodology is not clear and it can be further improved it.

(4)   In order to highlight the introduction, some latest references should be added to the paper for improving the reviews part. For example, 10.1016/j.isatra.2021.07.017 ;10.1109/JSTARS.2021.3059451 ; 10.3390/agriculture12060793 ï¼› 10.1007/s10489-022-03719-6 and so on.

(5)   The core technologies lack original innovation.

(6)   The comparison results with existing works should be proved by experiments rather than written description in Section 4.

(7)   Figure 9. is not clear. Please revise it.

(8)   The effectiveness of the proposed method needs to be verified by contrast experiments.

Round 2

Reviewer 2 Report (New Reviewer)

The manuscript was much improved, and still some issues should be well handled. For example, the reference is not enough for supporting the current paper.

line 191. references of radiometric calibration should be added for supporting here. I recommended some for this:

Radiometric Calibration for Multispectral Camera of Different Imaging Conditions Mounted on a UAV Platform

Radiometric calibration of UAV remote sensing image with spectral angle constraint

Relative radiometric calibration using tie points and optimal path selection for UAV images

Author Response

Reviewer 2

The manuscript was much improved, and still some issues should be well handled. For example, the reference is not enough for supporting the current paper.

line 191. references of radiometric calibration should be added for supporting here. I recommended some for this:

Radiometric Calibration for Multispectral Camera of Different Imaging Conditions Mounted on a UAV Platform

Radiometric calibration of UAV remote sensing image with spectral angle constraint

Relative radiometric calibration using tie points and optimal path selection for UAV images

The authors would like to thank the reviewer for providing the references to better support radiometric calibration. They were included in revised version of the manuscript.

Reviewer 3 Report (New Reviewer)

According to the revised paper, I have appreciated the deep revision of the contents and the present form of this manuscript.  There is little content, which need be revised according to the comment of reviewer in order to meet the requirements of publish. A number of concerns listed as follows:

(1) The authors need to interpret the meanings of the variables.

(2) How to determine these parameters? 

(3) Conclusion: What are the advantages and disadvantages of this study compared to the existing studies in this area?

(4)   How about the computation complexity of the proposed method?

(5) In order to further highlight the introduction, some latest references should be added to the paper for improving the reviews part.

Author Response

Reviewer 3

According to the revised paper, I have appreciated the deep revision of the contents and the present form of this manuscript.  There is little content, which need be revised according to the comment of reviewer in order to meet the requirements of publish. A number of concerns listed as follows:

REPLY: We would like to acknowledge the reviewer for helping us improving the manuscript with very pertinent comments and suggestions.

(1) The authors need to interpret the meanings of the variables.

REPLY: In the major revision phase, we did our best to adjust the manuscript to the reviewers' comments and suggestions, as recognized by all. Of course, we are willing to implement changes that can further improve the manuscript. With regard to this suggestion, if the reviewer is referring to section 2, all flight variables are fully explained and detailed. If the comment is meant to section 3, for each variable that characterizes an image or a table, there is also a detailed explanation in the respective captions. After another careful reading, we consider that all variables are fully explained. We did, although, add variables’ meaning to Figure 2 caption, to make it more explicit. However, if the reviewer continues to feel that there is some need to implement further detail regarding this topic, please let us know which ones, so that we can work on that.

(2) How to determine these parameters?

REPLY:  It is very difficult for the authors to identify which parameters is the reviewer referring to. Is it the flight parameters? If so, their determination falls outside this paper’s scope. However, they are (fully) presented as a characterization of each study itself. Acquired imagery is really what on point to this work. Flight parameters depend on the case study areas’ orography, UAV and sensor. Both the Portuguese and Italian case studies were conducted by experiment researchers very familiar with the available equipment and the regions. As such, parameters derive from that experience and from the manufacturer’s instructions (e.g. overlap, flight height, speed). In other case studies with the same or different equipment, aerial imagery acquisition could be done using other parameters. Therefore, besides detailing the process upon which aerial imagery was acquired for each case study – so that readers can know it and be able to repeat the process – the authors do consider that entailing the theoretical and experience basis used to determine flight parameters will further extend the paper without adding value to the reader.

(3) Conclusion: What are the advantages and disadvantages of this study compared to the existing studies in this area?

REPLY: This question is similar to one presented in the major revision, perhaps by a different reviewer. The difference of this study is the comparison of two very different hyperspectral sensors with regard to the overall preparation and data acquisition process over the same type of crop and under similar geo-environmental contexts. This work will hopefully provide many researchers, that are currently and in the near future looking for acquire hyperspectral equipment, the much-needed initial guide and early on support to be able to be aware of issues and advantages, but also to know procedures that enable them to develop R&D activities faster and easier. One of the main contributions is, in fact, the (detailed) description and implementation of hyperspectral imagery acquisition processes, as well as data processing and integration pipelines, when using two difference sensors, within precision viticulture context (in different countries). To the best of our knowledge, there are no other existing studies that can be compared to this one. With this, the authors are confident on having addressed the reviewer’s main concerns.

(4) How about the computation complexity of the proposed method?

REPLY: We would like to thank the reviewer for this question. There is no proposed method. Data computation steps were achieved by following different steps for each type of sensor. Section 3 presents this data with regard to processing time and mosaic size, which we hope should suffice to quantify and compare computation complexity for both pipelines.

(5) In order to further highlight the introduction, some latest references should be added to the paper for improving the reviews part.

REPLY: We understand the meaning of the comment and consider it relevant. Of course, it is always possible to go into more detail in the introduction, namely with regard to prior knowledge (background) to better understand the study. It seems to us that the introduction is well supported by references. However, a set of recent references has been included to improve this aspect.

This manuscript is a resubmission of an earlier submission. The following is a list of the peer review reports and author responses from that submission.

Round 1

Reviewer 1 Report

  • It's a very good report, very detailed, but I thought about it, what's the innovation?

  • I think some results could be  made without experimentation? 
    I dont know if you know this paper:Classification of imaging spectrometers for remote sensing applications, and I didn't get any more advanced information.Maybe we can focus on the impact on the application

Reviewer 2 Report

Dear authors,

thanks. Paper is now much more clear and understanable